# Transcriptomic Differences by RNA Sequencing for Evaluation of New Method for Long-Time In Vitro Culture of Cryopreserved Testicular Tissue for Oncologic Patients

**DOI:** 10.3390/cells13181539

**Published:** 2024-09-13

**Authors:** Cheng Pei, Plamen Todorov, Qingduo Kong, Mengyang Cao, Evgenia Isachenko, Gohar Rahimi, Frank Nawroth, Nina Mallmann-Gottschalk, Wensheng Liu, Volodimir Isachenko

**Affiliations:** 1Department of Obstetrics and Gynecology, Medical Faculty, Cologne University, 50931 Cologne, Germany; cheng.pei@uk-koeln.de (C.P.); qingduokong@gmail.com (Q.K.); mengyang.cao@uk-koeln.de (M.C.); evgenia.isachenko@uk-koeln.de (E.I.); gohar.rahimi@amedes-group.com (G.R.); nina.mallmann-gottschalk@uk-koeln.de (N.M.-G.); 2Institute of Biology and Immunology of Reproduction of Bulgarian Academy of Sciences (BAS), 1113 Sofia, Bulgaria; plamen.ivf@gmail.com; 3Medizinisches Versorgungszentrum AMEDES für IVF- und Pränatalmedizin in Köln GmbH, 50968 Cologne, Germany; 4Center for Infertility, Prenatal Medicine, Endocrinology and Osteology, Amedes Medical Center MVZ Hamburg, 20095 Hamburg, Germany; frank.nawroth@amedes-group.com; 5NHC Key Laboratory of Male Reproduction and Genetics, Guangdong Provincial Reproductive Science Institute (Guangdong Provincial Fertility Hospital), Guangzhou 510000, China; xwzliuwensheng@126.com

**Keywords:** human testicular tissue, cryopreservation, in vitro culture, fibrin granules, RNA sequencing, transcriptomics

## Abstract

Background: Earlier studies have established that culturing human ovarian tissue in a 3D system with a small amount of soluble Matrigel (a basement membrane protein) for 7 days in vitro increased gene fusion and alternative splicing events, cellular functions, and potentially impacted gene expression. However, this method was not suitable for in vitro culture of human testicular tissue. Objective: To test a new method for long-time in vitro culture of testicular fragments, thawed with two different regimes, with evaluation of transcriptomic differences by RNA sequencing. Methods: Testicular tissue samples were collected, cryopreserved (frozen and thawed), and evaluated immediately after thawing and following one week of in vitro culture. Before in vitro culture, tissue fragments were encapsulated in fibrin. Four experimental groups were formed. Group 1: tissue quickly thawed (in boiling water at 100 °C) and immediately evaluated. Group 2: tissue quickly thawed (in boiling water at 100 °C) and evaluated after one week of in vitro culture. Group 3: tissue slowly thawed (by a physiological temperature 37 °C) and immediately evaluated. Group 4: tissue slowly thawed (by a physiological temperature 37 °C) and evaluated after one week of in vitro culture. Results: There are the fewest differentially expressed genes in the comparison between Group 2 and Group 4. In this comparison, significantly up-regulated genes included C4B_2, LOC107987373, and GJA4, while significantly down-regulated genes included SULT1A4, FBLN2, and CCN2. Differential genes in cells of Group 2 were mainly enriched in KEGG: regulation of actin cytoskeleton, lysosome, proteoglycans in cancer, TGF-beta signaling pathway, focal adhesion, and endocytosis. These Group 2- genes were mainly enriched in GO: spermatogenesis, cilium movement, collagen fibril organization, cell differentiation, meiotic cell cycle, and flagellated spermatozoa motility. Conclusions: Encapsulation of testicular tissue in fibrin and long-time in vitro culture with constant stirring in a large volume of culture medium can reduce the impact of thawing methods on cryopreserved testicular tissue.

## 1. Introduction

The preservation of male fertility often involves cryopreservation of spermatozoa, a process wherein these spermatozoa can be obtained either through ejaculation or following a testicular biopsy. For prepubertal cancer patients lacking active spermatogenesis, traditional semen cryopreservation proves ineffective due to the absence of mature spermatozoa. Alternative methods, such as cryopreservation of spermatogonial stem cell-rich testicular cell or testicular tissue cryopreservation, become paramount in preserving fertility for these individuals.

In addition, while the enzymatic digestion of testicular tissue to isolate spermatogonial stem cells may compromise cell viability, cryopreservation of testicular tissue preserves crucial cell–cell and cell–matrix interactions [1,2]. The latter method emerges as potentially superior. Moreover, testicular tissue cryopreservation holds promise for adult patients grappling with genetic disorders or autoimmune diseases [3].

Testicular tissue in vitro culture stands as a pivotal method for unraveling the complexities of spermatogenesis and addressing male infertility concerns. Given the scarcity of human samples, research predominantly relies on animal models such as mice and pigs for studying in vitro culture and observing spermatogenesis dynamics [4,5]. It was reported that an attempt was made to culture in vitro cryopreserved human immature testicular tissues for 32 days, albeit without observing spermatogenesis [6].

The two primary approaches to in vitro culture of testicular tissue are the microfluidic-based dynamic culture system and the agarose-based static culture system [7]. Microfluidic devices crafted from polydimethylsiloxane (PDMS) hold promise for enhanced spermatogenesis efficiency owing to PDMS’s inherent affinity for biological samples [8]. However, the widespread adoption of this method faces challenges due to the demanding operational requirements of the device and the high cost of PDMS chips. In this study, drawing upon our team’s prior expertise with artificial ovaries, testicular tissue was encapsulated in TISSEEL Fibrin and subjected to in vitro culture [9].

RNA sequencing has significantly advanced understanding of testicular tissue development and pathology. Investigations performed on murine testicular tissue illuminate the dynamics of gene expression, across various stages of spermatogenesis. The minimal impact of cryopreservation on gene expression was shown [10].

We had the following reason for performing the experiments described in this article: we had earlier tested the protocol of in vitro culture of cryopreserved human ovarian tissue [11]. In short, cryopreservation (freezing and thawing) of tissue was performed by described protocol [12,13,14,15].

Experimental group pieces were placed on a floating membrane filter, after which a 3D culture system was created. To achieve this, Corning Matrigel Matrix, a soluble extract of basement membrane protein, was diluted with IVG medium (1:1), and 5 μL of the mixture was placed on the filter to form a small drop (one drop per membrane). Ovarian pieces were then placed on each drop (one piece per drop) and cultured for 7 days [11].

After 7 days of in vitro culture, the morphology of the ovarian tissue changed compared to un-cultured tissue. A dense fibrotic capsule formed, along with a slight reduction in volume [11]. The originally sharp edges of the ovarian cortical pieces became rounded, losing their angular shape. Additionally, cortical slices showed increased fibrosis, more follicular necrosis, and reduced living space for follicles, impacting the development of mature follicles [11]. At the DNA and RNA levels, gene fusion events occurred across nearly every chromosome, involving multiple genes and resulting in new patterns of gene expression and functional changes. These negative alterations [11] provided the basis for the investigations presented here.

Our investigations aimed to test the new method of testicular fragments thawed with two different regimes and long-time in vitro culture, with the evaluation of transcriptomic differences by RNA sequencing.

## 2. Materials and Methods

Unless specified otherwise, all chemicals were obtained from Sigma Chemical Co. (St. Louis, MO, USA).

### 2.1. Design of Experiments

Six testicular tissue samples were collected, cryopreserved (frozen and thawed) [12,13,14,15] and evaluated. Four groups were formed.

Group1: tissue quickly thawed (in boiling water at 100 °C) and immediately evaluated.

Group 2: tissue quickly thawed (in boiling water at 100 °C) and evaluated after one week of in vitro culture.

Group 3: tissue slowly thawed (by a physiological temperature 37 °C) and immediately evaluated.

Group 4: tissue slowly thawed (by a physiological temperature 37 °C) and evaluated after one week of in vitro culture.

Tree samples in each experimental group were used (Figure 1).

### 2.2. Collection of Samples

The experiments were performed in accordance with the Declaration of Helsinki and approved by the Bulgarian National Medical Institutional Ethics Committee (project “development of new cryopreservation methods to restore testicular function in adult and prepubertal patients with oncological diseases”, approval No. 7-021/2022) and the Institutional Ethics Committee of Cologne University (protocols 01-106, 12–163, 17-427, 20-1229, code BioMSOTE).

The technology tested in our studies is being developed for cancer patients. However, the testicular tissue fragments used for the experiments were obtained from patients involved in a fertility treatment program. Five patients were diagnosed with obstructive azoospermia; one patient was diagnosed with restricted azoospermia. Informed consent was obtained from patients whose tissue was collected for this study. Testicular tissue fragments (3 to 12 mm^3^) were obtained from six patients aged from 32 to 42 (median age 36.5 years). Two tissue fragments from each patient were used for experiments (a total of twelve tissue samples were collected for the research).

The procedure of extraction of testicular tissue has been previously described in detail [16,17,18]. A midline scrotal incision was performed, and the testis with the spermatic cord was removed, typically from the hemiscrotum containing the larger testis. The tunica vaginalis was opened to reveal the tunica albuginea, which was incised along the equatorial plane under a microscope, ensuring the preservation of subtunical vessels. After exposing the testicular parenchyma at 12× magnification, small tissue samples (3–10 mg) were excised by isolating thicker, opaque tubules from areas of Leydig cell nodules or hyperplasia.

### 2.3. Cryopreservation (Freezing and Thawing)

Testicular fragments were equilibrated for 30 min in cryovials containing a cryopreservation solution with 6% ethylene glycol (EG), 6% dimethyl sulfoxide (DMSO), and 0.15 M sucrose. The cryovials were then placed in an Ice Cube 14S freezer (SyLab, Neupurkersdorf, Austria) for conventional freezing, as described earlier for ovarian and testicular tissues [12,13,14,15]. The cryopreservation program consisted of the following stages: (1) initial cooling from −6 °C to −8 °C; (2) subsequent cooling from −6 °C to −34 °C at a rate of 0.3 °C/min; and (3) plunging the cryovials into liquid nitrogen at −34 °C. The protocol also included an auto-seeding step during the initial cooling phase. Subsequently, the cryovials were stored in liquid nitrogen for long-term preservation.

Tissues were thawed with two regimes:

*Quick thawing:* Cryovials were first exposed to room temperature for 30 s and then immersed in a 100 °C water bath for 60 s. The exposure time in boiling water was visually monitored using ice presence as a gauge. When ice reached 2-1mm, the cryovial was removed, resulting in a final medium temperature of 4 to 14 °C. Within 5 to 10 s after thawing, tissue fragments from the cryovials were transferred into a 10 mL thawing solution (basal medium with 0.5 M sucrose) in a 100 mL specimen container (Sarstedt, Nümbrecht, Germany). The tissues were exposed to the sucrose solution for 15 min, followed by a stepwise rehydration process, as previously described [12,13,14,15].

*Slow thawing:* Cryovials were first exposed to room temperature for 30 s and then immersed in a 37 °C water bath for 3 min. The subsequent thawing steps were identical to those used in the quick thawing regime described earlier.

### 2.4. 3D In Vitro Culture

The protocol for in vitro culture was previously used for the preparation of artificial ovaries [9].

TISSEEL Fibrin Sealant (Baxter International Inc., Deerfield, IL, USA) was employed for the encapsulation of testicular tissue for subsequent in vitro culture and analysis. The fibrin sealant comprises two main components: thrombin and fibrinogen, used at final concentrations of 10 IU/mL and 45.5 mg/mL, respectively, to achieve optimal gel formation and encapsulation. The two components were quickly mixed in an Eppendorf tube using a pipette and vortexed. Then 100 µL of solution was dropped onto testicular tissue to form a nearly gel-like mixture (Figure 2). Upon completion of gelation, the fibrin gel formed was gently peeled off using sterile tweezers to avoid any damages in tissue during in vitro culture.

The encapsulated tissue was then immediately transferred into a 700 mL cell culture flask (Greiner Bio-One GmbH, Frickenhausen, Germany) containing 70 mL of culture medium. The 7-day culture of testicular tissue was conducted in alpha-modified Eagle’s minimum essential medium (α-MEM, Life Technologies, Carlsbad, CA, USA) supplemented with 15% fetal calf serum, 0.1 mg/mL streptomycin, and 100 IU/mL penicillin. The cultures were maintained in a humidified incubator at 34 °C with 5% CO_2_ and agitated at 75 oscillations per minute using a rotating shaker.

### 2.5. Histology

Testicular tissues were fixed in 3.5% paraformaldehyde for 12 h at 4 °C and then embedded in paraffin wax. Sections with a thickness of 5 µm were cut, and every 10th section from each sample was mounted on a glass slide and stained with hematoxylin and eosin [19]. The section was subjected to morphological analysis of tissue development and viability under microscope Nikon Diaphot 300 (Nikon, Tokyo, Japan).

### 2.6. Data Extraction and Sequencing

Each testicular tissue sample underwent RNA extraction using the Trizol method. Subsequently, a strand-specific transcriptome library was created by isolating mRNA from total RNA. Sequencing was performed using the DNBSEQ high-throughput platform, followed by extensive bioinformatics analysis. RNA-seq data were analyzed using the Dr. Tom System (https://biosys.bgi.com accessed on 25 August 2023). Download the results of KEGG and GO related enrichment through this system. The raw RNA-seq data are accessible under BioProject: PRJNA1030294 and can be downloaded from the “Sequence read archive” on the National Center for Biotechnology Information (https://www.ncbi.nlm.nih.gov/bioproject/1030294 accessed on 14 March 2024).

## 3. Results

### 3.1. Morphology

In the cryopreservation process of testicular tissue presented in Figure 2(A2–C2), a halo (crown) was observed around the fragments. The process of removal of cryoprotectants occurred because of the dehydration of cells in 0.5 M sucrose when cryoprotectants were removed from them. The halo (crown) is explained by the different densities of sucrose and cryoprotectant solutions, which leave cells because of an osmotic reaction in cell membranes. It also illustrated the process wherein cryopreserved fragments of testicular tissue were sealed in fibrin granules before long-time in vitro culture (Figure 2(A4–C4)). Cryopreserved fragments of testicular tissue sealed in fibrin granules after long-time in vitro culture can be seen in Figure 3.

HE-staining of testicular tissue revealed improved morphology in specimens subjected to quick thawing followed by in vitro culture compared to quick thawing and immediate evaluation. In cells of Group 2 (quick thawing and in vitro culture) in comparison with Group 1 (quick thawing), the gap between the cells of the seminiferous tubules and the basement membrane was reduced, the basement membrane rupture was improved, and the nuclear condensation of spermatogonia was reduced (Figure 4(A1,A2,B1,B2)). Similarly, tissues after slow thawing followed by in vitro culture exhibited better morphology than those subjected to cells of group with slow thawing (Figure 4(C1,C2,D1,D2)). After in vitro culture, the contracted basement membrane was partially restored, the spermatogonia and supporting cells were plumper, and there were fewer gaps between adjacent seminiferous tubules (Figure 4). 

### 3.2. Differentially Expressed Genes

In cells of Group 2 (quick thawing and in vitro culture), 5513 up-regulated and 6422 down-regulated differential genes were detected in comparison with cells of Group 1 (quick thawing). Meanwhile, cells from Group 4 (slow thawing and in vitro culture) displayed 4533 genes with decreased expression and 3030 genes with increased expression compared to Group 3 cells (slow thawing). Comparison of Group 2 cells (quick thawing and in vitro culture) and Group 4 cells (slow thawing and in vitro culture) revealed 21 differential genes with increased expression and 11 genes with decreased expression in Group 2 (Figure 5A–C). In this comparison, significantly up-regulated genes included C4B_2, LOC107987373, and GJA4, while significantly down-regulated genes included SULT1A4, FBLN2, and CCN2.

Aggregate analysis of tissues thawed by both methods indicated that in cells from Group 2 (quick thawing and in vitro culture) and Group 4 (slow thawing and in vitro culture) 4818 up-regulated and 7660 down-regulated differential genes were detected in contrast with cells of Group 1 (quick thawing) and Group 3 (slow thawing). Also, it was established that up-regulated genes included UTP14C, GSTT1, and SCG2, and down-regulated genes included GAGE12E, GAGE12H, and CXorf51B (Figure 5D).

### 3.3. Enrichment Analysis of KEGG Pathways

Comparative KEGG pathway analysis revealed distinct patterns between cells from Group 2 (quick thawing and in vitro culture) and Group 1 (quick thawing), where the enriched pathways in Group 2 included lysosome, regulation of actin cytoskeleton, cellular senescence, MicroRNAs in cancer, and proteoglycans in cancer (Figure 6A).

Similarly, in comparison with cells of Group 3 (slow thawing), cells of Group 4 (slow thawing and in vitro culture) exhibited enrichment in pathways such as lysosome, focal adhesion, and calcium signaling pathway (Figure 6B). It was noted that the lysosome pathway showed the highest enrichment in both comparisons. In comparison with cells of Group 4, cells of Group 2 displayed enrichment in pathways like cytokine–cytokine receptor interaction, IL-17 signaling pathway, viral protein interaction with cytokine and cytokine receptor, Aldosterone synthesis and secretion, and protein digestion and absorption (Figure 6C). Comprehensive analysis across all groups revealed that the “thawing and in vitro culture” groups (Group 2 and Group 4) exhibited KEGG pathways, including regulation of actin cytoskeleton, lysosome, focal adhesion, TGF-beta signaling pathway, and proteoglycans in cancer, emphasizing the impact of in vitro culture on these biological processes (Figure 6D).

### 3.4. GO Biological Process Enrichment Analysis

In comparison with cells from Group 1 (quick thawing), cells from Group 2 (quick thawing and in vitro culture) predominantly exhibited enrichment in pathways related to spermatogenesis, flagellated sperm motility, cell projection organization, cilium movement, and meiotic cell cycle (Figure 6E). GO enrichment analysis of cells from Groups 3 and 4 primarily reflected pathways such as collagen fibril organization, cell adhesion, meiotic cell cycle, piRNA metabolic process, and extracellular matrix organization (Figure 6F).

In comparison with cells of Group 4, cells of Group 2 were predominantly enriched in pathways involving positive regulation of cell–substrate adhesion, positive regulation of cell differentiation, wound healing, positive regulation of axon regeneration, extracellular matrix organization, and spreading of cells (Figure 6G). It was noted that the different thawing methods after freezing may have a greater impact on the extracellular matrix during in vitro culture of testicular tissue (Figure 6H).

## 4. Discussion

### 4.1. Cryopreservation (Freezing and Thawing)

In our experiments, the protocols used for cryopreservation of testicular tissue were used [14], which were those previously used for ovarian tissue [9,12,13,15].

In our opinion, when we write about cryopreservation of ovarian and testicular tissue, the cells of both tissues can be presented as similar objects. In ovarian, as well as in testicular fragments, seven types of cell with similar intracellular structures can be differentiated. The main difference between testicular and ovarian tissues is their different density. The density of ovarian tissue is high, and testicular tissue has a loose structure. That is why, for in vitro culture of ovarian tissue fragments with a constantly stirred medium, there is no need to use encapsulation of this tissue. Whereas, due to the looseness of testicular tissue, after several days of large-volume in vitro culture with stirring of the medium, a fragment of such tissue would be dispersed into a suspension with small fragments. Therefore, we used fibrin capsules for culture of testicular tissue. However, the above differences in tissue density are insignificant in essence for cryopreservation: the cytoplasmic structures of cells of both tissues will be saturated with permeable cryoprotectants within minutes of the start of equilibration.

“Fresh” fragments (just after operation) were not an object of our experiments. This is because, for future clinical purposes, we will use only exactly cryopreserved (frozen and thawed) testicular tissues after anticancer therapy. In that way, the effectiveness of in vitro culture of “fresh” cells for our experiments is not necessary. We need to do in vitro culture of cells after cryopreservation to answer three questions regarding (1) general quality of the testicular tissue in this particular patient, (2) quality of testicular tissue in this particular patient after cryopreservation, and (3) how good is the whole cryopreservation process. The first question can be answered without in vitro culture, when a piece of testicular tissue is fixed and assessed immediately after surgery. In general, the aim of in vitro culture technology is to ensure that this technology maximally “copies” the parameters of the external (in situ) environment.

In our experiments, for thawing we used a 100 °C water bath. For the demonstration of this technological parameter, the following descriptions can be used: (1) when thawing in boiling water, a tampon was used to isolate the tissue fragment from the bottom of cryovial, which guaranteed no overheating; and (2) the temperature of the tissue fragment after thawing did not rise above 14 °C. For illustration, the cryovial felt cold 1 s after being removed from the boiling water. This technique was used because a tenet of classical cryobiology states that all types of cells and tissues, regardless of the cryopreservation method, must be thawed as quickly as possible.

### 4.2. 3D In Vitro Culture

The assessment of testicular tissue quality relies on comparing its condition before and after cryopreservation. The initial evaluation of fresh tissue offers insights into its viability post-cryopreservation, which is crucial for determining its utility. For instance, identifying pathological changes in fresh tissue indicative of impaired spermatogenesis renders thawing unnecessary. Subsequent examination post-thaw aims to gauge the efficacy of the freezing protocol in preserving cellular functionality essential for spermatogenesis.

An illustrative demonstration of the variance in cell quality post-thawing compared to subsequent culture arises in the context of ovarian tissue vitrification, a method of cryopreservation involving direct immersion in liquid nitrogen. Initially, the evaluation of follicle quality in such tissue immediately following thawing revealed a remarkable preservation state, virtually indistinguishable from that of freshly harvested follicles. This outcome aligned with expectations, given the use of permeable cryoprotectants at high concentrations, ensuring the follicles’ integrity during cryopreservation. However, within a mere six-hour culture period, nearly all follicles exhibited signs of degeneration. This striking discrepancy underscored the critical necessity for the development of dependable in vitro culture systems.

There was a previously proposed a culture system for ovarian tissue involving a large volume of culture medium with constant mixing. The efficacy of this approach with ovarian tissue fragments has been validated [20]. However, the use of the described system in relation to testicular tissue is impossible because human testicular tissue differs from ovarian tissue in density. Testicular tissue fragments tend to disintegrate into separate cell groups within 24 h in a dynamic system, impeding subsequent collection and analysis. Consequently, we devised a strategy involving the encapsulation of testicular fragments to prevent fragmentation during culture in a moving medium.

Fibrin, a substance widely present in mammalian organisms and approved for medical surgical practice by relevant regulatory organizations, such as the FDA and EU commissions, serves as a suitable encapsulating material. Its use may ensure the structural integrity of testicular tissue fragments during culture, facilitating subsequent analysis of cell quality post-culture.

The placement of tissue fragments on the periphery rather than at the center of fibrin granules, as depicted in Figure 2(A4–C4), aimed to enhance the diffusion of nutrients from the culture medium into the tissue. Examination of fibrin granules with tissue fragments after culture reveals that the tissues remain densely compacted and enveloped by a fibrin sheath (Figure 3).

After one week of in vitro culture by our method, a reduction in the volume of fibrin granule was observed. Nonetheless, tissue fragments remain ensconced in a fibrin layer (Figure 3(A5–C5)). In our experiments (unpublished data), instances were noted where, by the end of culture, a portion of the tissue (5–20% of the surface) was devoid of the fibrin layer (Figure 3(C4–C6,C8)). However, most of the tissue remained encapsulated in fibrin, and tissue dispersion did not occur during all period of in vitro culture.

### 4.3. RNA Sequencing

The primary role of testicular tissue lies in the production of spermatozoa. Extensive sequencing investigations in humans have elucidated that testicular tissue harbors a substantial repertoire of tissue-specific genes, distinguishing it as one of the most specialized tissues within the human body [21]. Current sequencing endeavors targeting human testicular tissue predominantly center on individuals afflicted with non-obstructive azoospermia, offering novel insights into potential biomarkers and underlying mechanisms [22,23,24,25]. In our study, a comprehensive analysis of transcriptomes derived from adult testicular tissues was conducted using RNA sequencing. Specifically, it examined the impact of various thawing methods and subsequent one week of in vitro culture of testicular tissues, aiming to deepen our understanding of the effects of the described method of in vitro culture.

#### 4.3.1. Differential Expressed Genes

In the present study, comparison of cells from Group 2 (quick thawing and in vitro culture) and Group 4 (slow thawing and in vitro culture) revealed a comparatively limited number of differentially expressed genes, totaling only 32. The most conspicuous up-regulation was C4B_2, while SULT1A4 exhibited the most pronounced down-regulation.

C4B_2, formally known as Complement Component 4B (Chido Blood Group), Copy 2, is identified as a protein-binding gene. Its heightened expression has been observed in patients with familial primary myelofibrosis, suggesting its potential involvement in extracellular matrix formation and local inflammatory responses [26]. This observation aligns with our findings, indicating that elevated expression of C4B_2 may promote extracellular matrix development in testicular tissues during in vitro culture. Conversely, diminished expression of C4B_2 has been recognized as a risk factor for systemic lupus erythematosus, an autoimmune condition [27,28].

Turning to SULT1A4, or Sulfotransferase Family 1A Member 4, it is a pivotal enzyme in catecholamine metabolism [29]. It is noteworthy that SULT1A4 shares an important paralogous relationship with SULT1A3, and several studies have posited a potential association between the copy number of SULT1A3/4 and neurodegenerative diseases such as Parkinson’s and Alzheimer’s [30]. Furthermore, recent investigations suggest that SULT1A4 may emerge as a significant target of di(2-ethylhexyl) phthalate in human granulosa cells, a compound often linked to adverse effects on female reproduction [31].

#### 4.3.2. KEGG Pathways Analysis

The KEGG enrichment analysis of Groups 2 and 4 genes, characterized by a limited number of differential genes, underscored a notable enrichment in the cytokine–cytokine receptor interaction pathway. As soluble proteins that circulate outside cells, cytokines produce their effects by attaching to specific receptors located on the surface of target cells. Disruptions in cytokine–cytokine receptor interaction within the testicular milieu have been associated with various male reproductive disorders [32,33]. Additionally, cytokines play pivotal roles in the immune responses and inflammation within testicular tissue. Furthermore, comparing these two groups revealed a significant involvement of the IL-17 signaling pathway. The IL-17 cytokine family includes six members, IL-17A to F, with IL-17A exhibiting pronounced signaling pathway characteristics [34]. IL-17A is often implicated in the recruitment of inflammatory cells into the testicular interstitium, thereby compromising the integrity of the blood–testis barrier and subsequently impacting seminiferous tubule function [35]. Notably, the knockdown of IL-17A has demonstrated efficacy in mitigating testicular immune responses induced by factors such as fluoride exposure, thereby ameliorating spermatogenic damage [36]. Additionally, supplementation with probiotics has shown promise in attenuating IL-17A signaling activation driven by intestinal microbiota, consequently mitigating inflammation-induced declines in spermatozoon quality [37].

Except for cells of Groups 2 and 4, KEGG pathway analysis revealed that the lysosomal pathway exhibited the most significant changes in the remaining groups. Lysosomes, besides their role in cellular waste processing via autophagy, endocytosis, and phagocytosis pathways, play crucial roles in immune cell signaling, nutrient sensing, and metabolism [38]. During spermatogenesis, a significant proportion of testicular germ cells undergo apoptosis, which is subsequently engulfed and degraded by testicular Sertoli cells [39]. Processes such as the development of haploid germ cells, degradation of spermatozoon cytoplasm, and provision of energy for spermatozoon motility are intricately associated with autophagy [40]. The lysosomal pathway suggests extensive metabolic activity and spermatogenesis within testicular tissues during in vitro culture.

Additionally, there is significant enrichment observed in the regulation of actin cytoskeleton. The actin cytoskeleton plays pivotal roles across various stages of spermatogenesis, facilitating attachment in different cell junctions, such as basal and apical ectoplasmic specializations within the blood–testis barrier [41,42]. Basal ectoplasmic specializations regulate the transport of preleptotene spermatocytes, while the degradation of apical ectoplasmic specializations ensures the entry of fully developed spermatozoa into the lumen [43]. The dynamic renewal and homeostasis of actin are vital for spermatozoa development.

#### 4.3.3. GO Terms Analysis

Cells from Group 2 (quick thawing and in vitro culture) and Group 4 (slow thawing and in vitro culture) demonstrate positive regulation of cell–substrate adhesion emerged as the top-ranked GO enrichment. Alterations in integrin attachment often drive changes in cell–substrate adhesion, consequently influencing cell migration and development. β1-integrin, along with its associated kinase integrin-linked kinase (ILK), plays pivotal roles in mediating adhesion between Sertoli cells and spermatids during spermatogenesis [44]. Furthermore, spermatid releases are mediated by a “detachment complex” containing phosphorylated FAK and α6β1-integrin [45]. Additionally, the combined GO enrichment analysis in cells from remaining groups highlighted biological processes such as spermatogenesis, cilia motility, and cell differentiation as prominently enriched, attributed to in vitro culture. GO analysis revealed distinctions between Group 3 (slow thawing) and Group 4 (slow thawing and in vitro culture) cells. In this case, cell adhesion and collagen fiber organization are more pronounced.

In our experiments, RNA sequencing of testicular tissue and testicular germ cells was performed, and somatic cells were not identified. This can only provide a general understanding of the KEGG and GO enrichment of differentially expressed genes in testicular tissue before and after in vitro culture. The adult spermatozoon maturation cycle is about 90 days but, in this study, tissue was cultured in vitro for only 7 days. The culture time is short, and only part of the stages of spermatozoon maturation can be observed.

## 5. Conclusions

Encapsulation of testicular tissue in fibrin and long-time in vitro culture with constant stirring in a large volume of culture medium can reduce the impact of thawing methods on cryopreserved testicular tissue.

## Figures and Tables

**Figure 1 cells-13-01539-f001:**
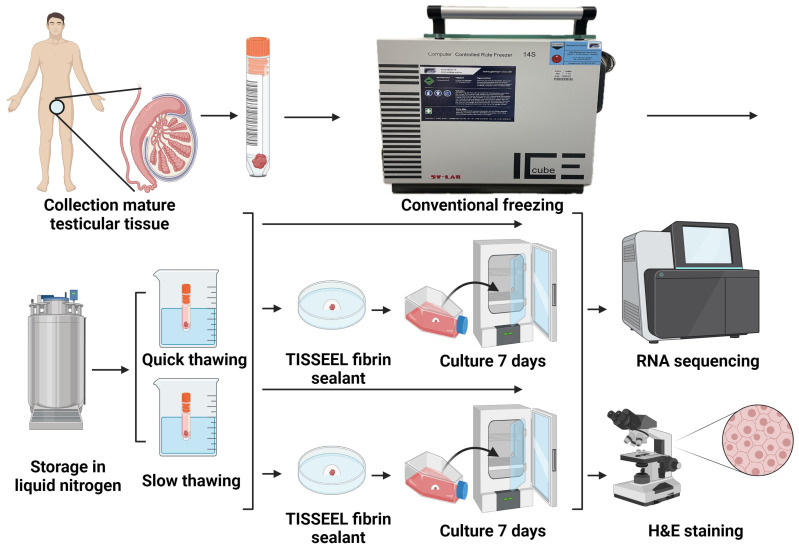
Design of experiments.

**Figure 2 cells-13-01539-f002:**
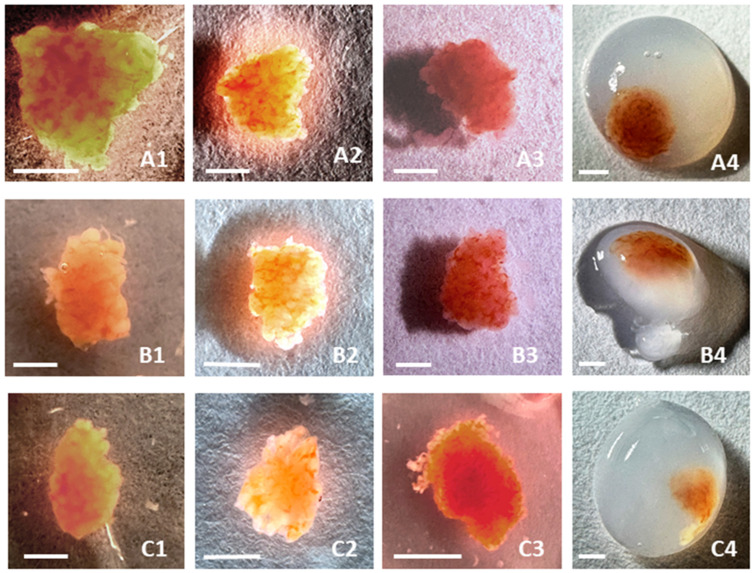
Cryopreserved fragments of testicular tissue from three patients. (**A1**–**C1**) Cryopreserved tissue fragments from three patients immediately after thawing (at 100 °C) in a freezing solution (6% dimethyl sulfoxide + 6% ethylene glycol + 0.15 M sucrose). (**A2**–**C2**) The same fragments 3 min after the beginning of removal of cryoprotectants in 0.5 M sucrose. (**A3**–**C3**) The same fragments in an isotonic solution after the end of removal of cryoprotectants (rehydration) (**A4**–**C4**) The same fragments 1 min after the beginning of formation of fibrin granules. Bar = 1.0mm.

**Figure 3 cells-13-01539-f003:**
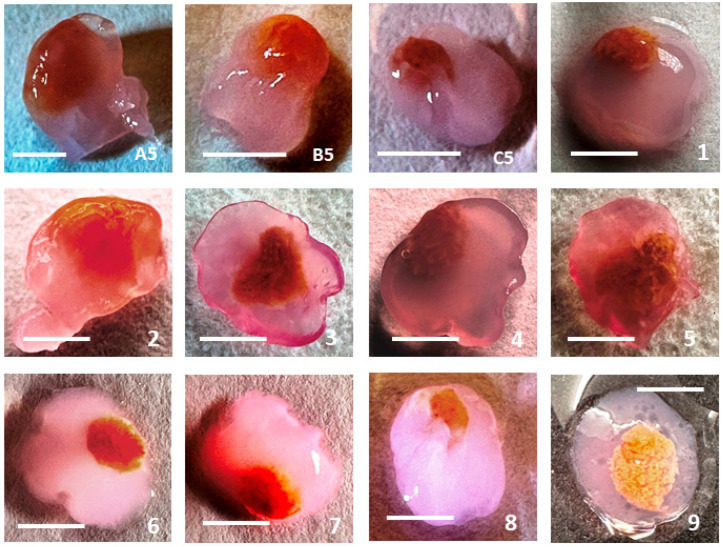
Cryopreserved fragments of testicular tissue sealed in fibrin granules after long-time in vitro culture. (**A5**–**C5**) Tissue fragments from three patients A, B, and C, shown in Figure 2 (**1**–**8**). Photos show the “behavior” of fibrin granules with fragments embedded using various embedding methods after long-time in vitro culture, (unpublished data). (**9**) The process of embedding a testicular tissue fragment in a fibrin gel: photo demonstrating the friability of the fragment and, consequently, the inevitability of its disintegration during in vitro culture. Bar = 2.0 mm.

**Figure 4 cells-13-01539-f004:**
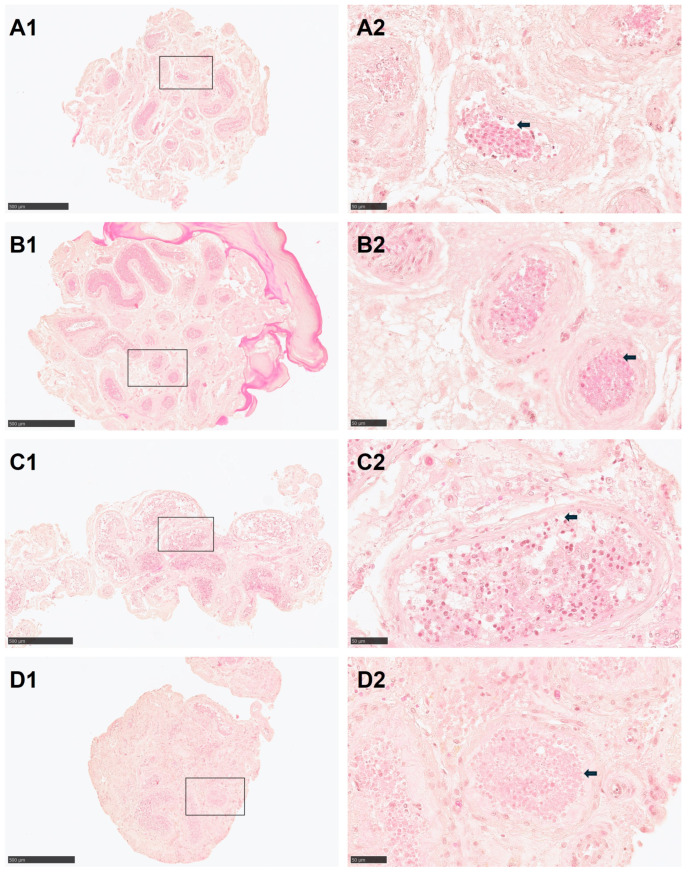
Hematoxylin-Eosin (HE)-staining of cryopreserved and in vitro cultured testicular tissue. (**A1**,**A2**) HE-staining of cells from Group 1 (quick thawing). (**B1**,**B2**) HE-staining of Group 2 (quick thawing and in vitro culture). (**C1**,**C2**) HE-staining of Group 3 (slow thawing). (**D1**,**D2**) HE-staining of Group 4 (slow thawing and in vitro culture). Bar for (**A1**–**D1**) = 500 μm, Bar for (**A2**–**D2**) = 50 μm. Black arrow indicates the space between basal membrane and cells in seminiferous tubules.

**Figure 5 cells-13-01539-f005:**
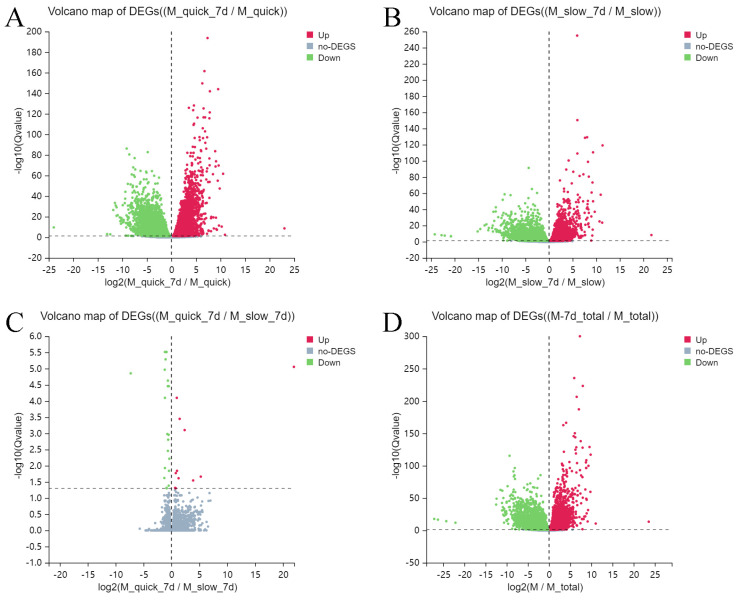
Volcano map showing differentially expressed genes (DEGs) between cryopreserved and in vitro cultured testicular tissue. (**A**) DEG volcano map: Group 2 cells (quick thawing and in vitro culture) vs. Group 1 (quick thawing). (**B**) DEG volcano map: Group 4 (slow thawing and in vitro culture) vs. group 3 (slow thawing). (**C**) DEG volcano map: Group 2 (quick thawing and in vitro culture) vs. Group 4 (slow thawing and in vitro culture). (**D**) DEG volcano map: Group 2 (quick thawing and in vitro culture) and Group 4 (slow thawing and in vitro culture) vs. Group 1 (quick thawing) and Group 3 (slow thawing).

**Figure 6 cells-13-01539-f006:**
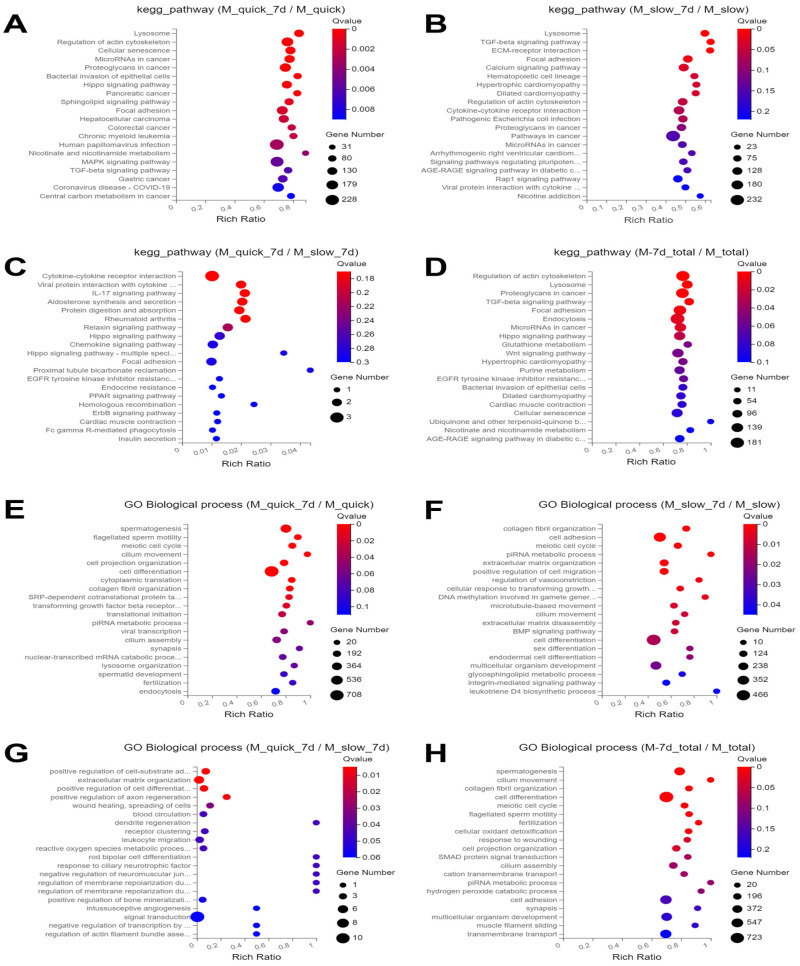
Bubble chart of differentially expressed genes (DEGs) displaying KEGG pathways and GO enrichment. (**A**) KEGG pathway chart from Group 2 (quick thawing and in vitro culture) and from Group 1 (quick thawing). (**B**) KEGG pathway chart of Group 4 cells (slow thawing and in vitro culture) and Group 3 cells (slow thawing). (**C**) KEGG pathway chart from Group 2 cells (quick thawing and in vitro culture) and from Group 4 (slow thawing and in vitro culture). (**D**) KEGG pathway chart of DEG in cells from Group 2 (quick thawing and in vitro culture) and Group 4 (slow thawing and in vitro culture) vs. cells of Group 1 (quick thawing) and Group 3 (slow thawing). (**E**) GO enrichment bubble chart for cells from Group 2 (quick thawing and in vitro culture) and Group 1 (quick thawing). (**F**) GO enrichment bubble chart for cells from Group 4 (slow thawing and in vitro culture) and Group 3 (slow thawing). (**G**) GO enrichment bubble chart for Group 2 (quick thawing and in vitro culture) and Group 4 (slow thawing and in vitro culture). (**H**) GO enrichment bubble chart for Group 2 (quick thawing and in vitro culture) and Group 4 (slow thawing and in vitro culture) vs. Group 1 (quick thawing) and Group 3 (slow thawing).

## Data Availability

The raw RNA-seq data are accessible under BioProject: PRJNA1030294 and can be downloaded from the “Sequence read archive” on the National Center for Biotechnology Information (https://www.ncbi.nlm.nih.gov/bioproject/1030294 accessed on 14 March 2024).

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
