# Peer review of "Transcriptomic Differences by RNA Sequencing for Evaluation of New Method for Long-Time In Vitro Culture of Cryopreserved Testicular Tissue for Oncologic Patients"

_cells, 2024, doi:10.3390/cells13181539_

Round 1

Reviewer 1 Report

Comments and Suggestions for Authors

This is a very interesting manuscript related to the transcriptomic differences by RNA sequencing for evaluation of new method for long-time in vitro culture of cryopreserved testicular tissue in oncologic patients. Manuscript is very well written, but some issues were detected and reported below:

1. With regards to samples collection, from lines 107 to 116, the history of the patients that donate the testicular tissues is not clear. Were they previously fertile men that lost the fertility due to cancer or chemotherapy? Were they always azoospermic? If all of them were oncologic patient, what kind of cancer they present? Please clarify this.

2. For curiosity only, how was the recovery of the patients after testicular tissues extraction?

3. Please provide a few more information related to the method used for freezing testicular tissues. Provide data related to the time and temperatures adopted for the freezing curve? Could such method be considered a slow freezing? 

4. Please provide the size of the testicular tissue fragments used.

5. What is the origin of the thawing methods used? Why did you decided to test 100 or 37 degrees? Is there any previous report for the use of a high thawing temperature as 100 degrees for testicular or other tissue? Please reference your choices or provide your hypothesis for using such temperature at the introduction section.

6. Reference slow thawing, since this is the most used method in general.

7. Please specify the type of culture you conducted. It seems to be a 3D culture, not an organotipic one, isn't it?

8. Please, provide references for all your methods, including histology.

9.  Since you conducted histological evaluation, why not analyzing scores related to tubular morphology, as cell loss, membrane integrity, lumen presence, tubule architecture, swelling, etc? I was curious about the relationship of conventional histological results and the molecular approach you conducted.

10. Were all data only evaluated on a subjective way? Couldn't the results related to the number of up and down regulated genes for each group be compared by using and appropriate statistical test to exclude the subjectivity in such analysis? If any statistical analysis was conducted, please provide the descriptions in methodology section.

11. The second paragraph in discussion section lacks a reference, specially for data related to ovarian tissue.

12. Regarding discussions related to in vitro culture, authors should assume that this was only used as an analysis method, since literature demonstrate that for a testicular tissue IVC to really obtain the in vitro spermatogenesis, a very long culture (from 2 to 3 months) are necessary. So, authors are advised to assume the limitations of their work.

13. Regarding the differential expression of genes, I confess that I am not sure if the results obtained were bad or good. If genes C4B_2 and SULT1A4 are associated to some lupus and degenerative diseases, wouldn't their up regulation in groups 2 and 4 a deleterious factor?

14. On the discussion related to the KEGG, authors highlight the notable enrichment of citokyne receptors in testicular tissues, specially in the cultured groups. How would this be a positive result since the self authors explain the involvement of cytokines and inflammatory process?? 

15. I did not understand the reason why authors describe previous experiments in ovarian tissue from lines 416 to 443. If such information were used to explain the present data ok, but it is not. So, I believe that such description is unnecessary.

16. Based on my comments, I am really not sure if the authors conclusions are the most adequate. After revising discussion, please, revise the conclusions. 

Author Response

Reviewer 1.

 This is a very interesting manuscript related to the transcriptomic differences by RNA sequencing for evaluation of new method for long-time in vitro culture of cryopreserved testicular tissue in oncologic patients. Manuscript is very well written, but some issues were detected and reported below:

  1. With regards to samples collection, from lines 107 to 116, the history of the patients that donate the testicular tissues is not clear. Were they previously fertile men that lost the fertility due to cancer or chemotherapy? Were they always azoospermic? If all of them were oncologic patient, what kind of cancer they present? Please clarify this.

Autors.

Dear Sir/Madam,

Thank you very much for your careful work with our manuscript. We accepte all your remarks and respective changes were introduced in the text. Now we have written:

The technology being tested in our studies is being developed for cancer patients.

However, the testicular tissue fragments used for the experiments were obtained from patients from a fertility treatment program. Five patients were diagnosed with obstructive azoospermia, one patient was diagnosed with restricted azoospermia. 

Reviewer 1. 2. For curiosity only, how was the recovery of the patients after testicular tissues extraction?

Authors.Thank you for your question. As we have written in manuscript now, these patients have obtained reproductive treatment (IVF-procedure). We have used testicular tissue of sterile (azzoospermic) patients for experiments for technology for cancer patients. In our cryobank we have also frozen testicular fragments obtained before anticancer treatment but we have not concern from these patients. To the present three patients have children born after using of testicular tissue for IVF-procedure. Two patients have failed to conceive, and one patient has not yet conceived using assisted reproductive technology. As I have noted above, it is written now:

The technology being tested in our studies is being developed for cancer patients.

However, the testicular tissue fragments used for the experiments were obtained from patients from a fertility treatment program. Five patients were diagnosed with obstructive azoospermia, one patient was diagnosed with restricted azoospermia.

Also we have written:

Testicular tissue fragments (3 to 4 mm3) was obtained from 6 patients aged from 32 to 42 (median age 36,5 years). Two tissue fragments from each patient were used for experiments (total 12 tissue samples were collected for research. 

The procedure of extraction of testicular tissue has been previously described in detail [14-16]. Briefly, a midline incision was made in the scrotum and the testis, and the spermatic cord was removed, preferably from the hemiscrotum, with the larger testis. Tunica vaginalis was opened, and Tunica albuginea was visualized. Under an operating microscope, Tunica albuginea was widely opened in an equatorial plane, preserving the subtunical vessels. After the opening of Tunica albuginea, testicular parenchyma was examined directly at 12-fold magnification under the operating microscope. Small samples (9–20 mg) were excised by pulling out larger, more opaque tubules from surrounding Leydig cell nodules or hyperplasia in the testicular parenchyma.

Reviewer 1. 3. Please provide a few more information related to the method used for freezing testicular tissues. Provide data related to the time and temperatures adopted for the freezing curve? Could such method be considered a slow freezing? 

Autors.

Dear Sir/Madam,

Your remark is accepted and now we have written the following:

The cryopreservation program included the following stages: (1) starting temperature was −6°C to -8°C; (2) samples were cooled from −6°C to −34°C at a rate of 0.3 â—¦C/min; and (3) at −34 â—¦C cryo-vials were plunged into liquid nitrogen. The freezing protocol for cryopreservation of this ovarian tissue included an auto-seeding step at -6°C to -8°C.

Taking into account your recommendation to describe our cryopreservation methods, we have added to Discussion the following:

Fragments just after operation (“fresh”) were not presented in experiments because for experiments we have used only cryopreserved (frozen and thawed) testicular tissues. In that way the effectiveness of in vitro culture of "fresh" cells is not actual. We need to do in vitro culture of cells after cryopreservation to answer two questions:

  1. What is the quality of the testicular tissue in this particular patient in general?
  2. What is the quality of testicular tissue in this particular patient after cryopreservation?
  3. How good is the whole cryopreservation process?

We can answer the first question without in vitro culture, when a piece of testicular tissue is fixed and assessed immediately after surgery. In this case, we can see the morphology of this tissue.

But the fact is that for future manipulations with tissue we will have only cryopreserved tissue that will be used immediately after thawing, without in vitro culture for one week.

The aim of in vitro culture technology is to ensure that this technology maximally "copies" the parameters of the external environment.  

In our experiments, for thawing we have used 100°C water bath. For demonstration of this technological parameter the following descriptions can be used: 1) when thawing in boiling water, it is used a tampon that isolates the tissue fragment from the bottom of cryovial, which guarantees no overheating; 2) the temperature of the tissue fragment after thawing does not rise above 14°C. For illustration: the cryo-vial after being removed from the boiling water after 1 sec. feels cold. This technique is used because a tenet of classical cryobiology states that all types of cells and tissues, regardless of cryopreservation method, must be thawed as quickly as possible.

Reviewer 1. 4. Please provide the size of the testicular tissue fragments used.

Autors.

Dear Sir/Madam,

Your remark is accepted and now we have written the following:

Testicular tissue fragments (3 to 4 mm3) was obtained from 6 patients aged from 32 to 42 (median age 36,5 years). Two tissue fragments from each patient were used for experiments (total 12 tissue samples were collected for research). 

Reviewer 1. 5. What is the origin of the thawing methods used? Why did you decided to test 100 or 37 degrees? Is there any previous report for the use of a high thawing temperature as 100 degrees for testicular or other tissue? Please reference your choices or provide your hypothesis for using such temperature at the introduction section.

Autors.

Dear Sir/Madam,

Your remark is accepted and now we have written the following:

In our experiments, for thawing we have used 100°C water bath. For demonstration of this technological parameter the following descriptions can be used: 1) when thawing in boiling water, it is used a tampon that isolates the tissue fragment from the bottom of cryovial, which guarantees no overheating; 2) the temperature of the tissue fragment after thawing does not rise above 14°C. For illustration: the cryo-vial after being removed from the boiling water after 1 sec. feels cold. This technique is used because a tenet of classical cryobiology states that all types of cells and tissues, regardless of cryopreservation method, must be thawed as quickly as possible.

Reviewer 1. 7. Please specify the type of culture you conducted. It seems to be a 3D culture, not an organotipic one, isn't it?

Autors.

Dear Sir/Madam,

You are right and thank you for your suggestion. Your remark is accepted and now we have written the following: 3D in vitro culture.

Reviewer 1. 8. Please, provide references for all your methods, including histology.

And also

Reviewer 1. 9.  Since you conducted histological evaluation, why not analyzing scores related to tubular morphology, as cell loss, membrane integrity, lumen presence, tubule architecture, swelling, etc? I was curious about the relationship of conventional histological results and the molecular approach you conducted.

Autors.

Dear Sir/Madam,

Both your remark is accepted and now we have written the following:

 HE-staining of testicular tissue revealed improved morphology in specimens subjected to quick thawing followed by in vitro culture in comparison with quick thawing and immediate evaluation. In cells of Group 2 (quick thawing + in vitro culture) in comparison with cells of Group 1 (quick thawing), the gap between the basement membrane and cells of the seminiferous tubules was reduced. The basement membrane rupture was improved, and the nuclear condensation of spermatogonia was reduced (Fig.4, A1, A2, B1, B2). Similarly, tissues after slow thawing followed by in vitro culture exhibited better morphology than those subjected to slow thawing group (Fig. 4, C1, C2, D1, D2). After in vitro culture, the contracted basement membrane was partially restored, the spermatogonia and supporting cells were plumper, and there were fewer gaps between adjacent seminiferous tubules (Fig.4).

Reviewer 1. 10. Were all data only evaluated on a subjective way? Couldn't the results related to the number of up and down regulated genes for each group be compared by using and appropriate statistical test to exclude the subjectivity in such analysis? If any statistical analysis was conducted, please provide the descriptions in methodology section.

Authors:

Thank you very much for your suggestion. The results of differentially expressed genes, KEGG pathway analysis, and GO enrichment analysis are all derived from the objective data obtained from RNA sequencing analysis of the collected testicular tissues. These statistical data have been rigorously processed and analyzed to ensure the reliability and scientificity of the results. Our RNA sequencing analysis is provided by the Dr. Tom System (https://biosys.bgi.com), which is an advanced bioinformatics platform dedicated to the processing and analysis of high-throughput sequencing data.

For example, the results section states:

In cells of Group 2 (quick thawing + in vitro culture) it was detected 5513 up-regulated and 6422 down-regulated differential genes in comparison with cells of Group 1 (quick thawing).

Comparison of Group 2 cells (quick thawing + in vitro culture) and cells of Group 4 (slow thawing + in vitro culture) revealed in Group 2 twenty-one differential genes with increased expression and 11 genes with decreased expression.

Reviewer 1. 11. The second paragraph in discussion section lacks a reference, specially for data related to ovarian tissue.

Autors.

Dear Sir/Madam,

Your remark is accepted and now we have added respective references (18, 20).

Reviewer 1. 12. Regarding discussions related to in vitro culture, authors should assume that this was only used as an analysis method, since literature demonstrate that for a testicular tissue IVC to really obtain the in vitro spermatogenesis, a very long culture (from 2 to 3 months) are necessary. So, authors are advised to assume the limitations of their work.

Autors.

Dear Sir/Madam,

Your remark is accepted and now we have written the following:

Authors note that one-week in vitro culture was used only as an analysis method, because for in vitro obtaining of spermatogenesis, a very long culture (from 2 to 3 months) is necessary. This is limitations of our work.

Reviewer 1. 13. Regarding the differential expression of genes, I confess that I am not sure if the results obtained were bad or good. If genes C4B_2 and SULT1A4 are associated to some lupus and degenerative diseases, wouldn't their up regulation in groups 2 and 4 a deleterious factor?

Autors.

Thank you very much for your suggestion.

In the comparison of differentially expressed genes between group 2 and group 4, C4B_2 is the most obviously upregulated differentially expressed gene. The upregulated expression of this gene is often related to the formation of extracellular matrix, while the reduced expression of this gene has been identified as a risk factor for systemic lupus erythematosus. SULT1A4 is the most obviously downregulated differentially expressed gene. The high expression of this gene is often related to neurodegenerative diseases and may have adverse effects on the reproductive system.

Therefore, C4B_2 and SULT1A4 are not related. They are just the most obviously upregulated and downregulated genes in these two groups of differentially expressed genes.

Reviewer 1. 14. On the discussion related to the KEGG, authors highlight the notable enrichment of citokyne receptors in testicular tissues, specially in the cultured groups. How would this be a positive result since the self authors explain the involvement of cytokines and inflammatory process?? 

Autors.

Dear Sir/Madam,

It is noted in the text now: IL-17A is often implicated in the recruitment of inflammatory cells into the testicular interstitium, thereby compromising the integrity of the blood-testis barrier and subsequently impacting seminiferous tubule function [36]. Notably, the knockdown of IL-17A has demonstrated efficacy in mitigating testicular immune responses induced by factors such as fluoride exposure, thereby ameliorating spermatogenic damage [37].

Because the IL-17 cytokine family is closely related to testicular tissue development and spermatogenesis, it can be inferred that the IL-17 signaling pathway plays an important role in testicular tissue development and spermatogenesis after cryopreservation. The results of KEGG enrichment analysis of groups 2 and 4 showed that the enrichment level of this pathway was high, which further indicated that the rapid warming plus in vitro culture scheme had a greater impact on the IL-17 signaling pathway. It suggests that the method of quick thawing combined with in vitro culture may promote the recovery of testicular tissue and spermatogenesis after cryopreservation by regulating the IL-17 signaling pathway.

Reviewer 1. 15. I did not understand the reason why authors describe previous experiments in ovarian tissue from lines 416 to 443. If such information were used to explain the present data ok, but it is not. So, I believe that such description is unnecessary.

Autors.

Dear Sir/Madam,

Thank you for your question.

I can explain why we described this method with details.

Fact is that we have tried this method with small volume of a regularly changed culture medium for in vitro culture of human ovarian pieces. This is method which described in literature as very promising.

Our results shown that it is bad. The following our attempt is to test our old method which we used many times (and published) for ovarian tissue: large volume of culture medium (you are right, 3D is suitable word) with fortexing. 

We ask you to be agree to leave this fragment in the manuscript.

Again thank you for your work. 

Reviewer 2 Report

Comments and Suggestions for Authors

I've found some aspects that deserve to be revised. Please improve the discussion of the in vitro culture method used by the authors. 

From all of differentiated genes there were only two of interest. Please describe why those two were so relevant.  

Comments on the Quality of English Language

Please check for minor errors.

Author Response

Reviewer 2.

I've found some aspects that deserve to be revised. Please improve the discussion of the in vitro culture method used by the authors. 

Authors.

Dear Sir/Madam,

Thank you very much for your careful work with our manuscript. We accepte all your remarks and respective changes were introduced in the text. Now we have corrected the description of method.

Reviewer 2. From all of differentiated genes there were only two of interest. Please describe why those two were so relevant. 

Autors.

Dear Sir/Madam,

Thank you very much for your suggestion.

Among the four groups, only group 2 and group 4 had the least number of differentially expressed genes. Specifically, only 21 differentially expressed genes showed upregulation, and another 11 differentially expressed genes showed downregulation. In contrast, the number of differentially expressed genes in other groups increased significantly, reaching thousands. Therefore, the focus of the study was on the comparative analysis of differentially expressed genes between group 2 and group 4.

In these two groups, the upregulation of the C4B_2 gene was the most significant, while the downregulation of the SULT1A4 gene was the most obvious. The significant changes in the expression levels of these two genes suggest that they may play an important role in testicular tissue, so the discussion section focuses on the functions of the C4B_2 and SULT1A4 genes and their potential effects on testicular tissue.

Reviewer 3 Report

Comments and Suggestions for Authors

The manuscript is presenting an essential subject that important in the cryopreservation of testicular tissues - freezing and thawing testicular tissues.  However, the results did not contain cardinal experiments which are valuable for male fertility preservation.

1. The authors did not provide a table that summarizes clinical values for each patient such as age, hormone levels, type of azoospermai, size of the testis, and others.

2. The results are missing the histology of the biopsies before freezing.

3. The authors did not identify the type of somatic cells and the types of spermatogenic cells present before and after freezing and after thawing and culture. This is important to show the effect of the different protocols on cells that regulate and are involved in spermatogenesis.

4. The reviewer suggests to the authors to present data of correlation between RNA seq and the presence/activity of testicular somatic cells and also with the different types of spermatogenic cells present in both protocols.

missing some 

Author Response

Reviewer 3. 1. The authors did not provide a table that summarizes clinical values for each patient such as age, hormone levels, type of azoospermai, size of the testis, and others.

Autors.

Dear Sir/Madam,

Your remark is accepted and now we have written the following:

The technology being tested in our studies is being developed for cancer patients. However, the testicular tissue fragments used for the experiments were obtained from patients from a fertility treatment program. Five patients were diagnosed with obstructive azoospermia, one patient was diagnosed with restricted azoospermia. The informed consent was obtained from patients whose tissue was collected for this study. Testicular tissue fragments (3 to 4 mm3) was obtained from 6 patients aged from 32 to 42 (median age 36,5 years). Two tissue fragments from each patient were used for experiments (total 12 tissue samples were collected for research. 

Reviewer 3. 2. The results are missing the histology of the biopsies before freezing.

Autors.

Dear Sir/Madam, Your remark is accepted and now we have written in discussion the following explanation why the object "fresh testicular tissue" is not the object of our research. It is because in the future, some years after anticancer therapy, when for sure testicular tissues will be death, we have only one possibility, to use only thawed testicular tissue.

We have written:

Fragments just after operation (“fresh”) were not presented in experiments because for experiments we have used only cryopreserved (frozen and thawed) testicular tissues. In that way the effectiveness of in vitro culture of "fresh" cells is not actual. We need to do in vitro culture of cells after cryopreservation to answer two questions:

  1. What is the quality of the testicular tissue in this particular patient in general?
  2. What is the quality of testicular tissue in this particular patient after cryopreservation?
  3. How good is the whole cryopreservation process?

We can answer the first question without in vitro culture, when a piece of testicular tissue is fixed and assessed immediately after surgery. In this case, we can see the morphology of this tissue.

The aim of in vitro culture technology is to ensure that this technology maximally "copies" the parameters of the external environment. 

Reviewer 3. 3. The authors did not identify the type of somatic cells and the types of spermatogenic cells present before and after freezing and after thawing and culture. This is important to show the effect of the different protocols on cells that regulate and are involved in spermatogenesis.

  1. The reviewer suggests to the authors to present data of correlation between RNA seq and the presence/activity of testicular somatic cells and also with the different types of spermatogenic cells present in both protocols.

Autors. Thank you very much for your suggestion. This manuscript only conducted a global RNA sequencing study on different testicular tissues, and the scope of the study is relatively general. If we want to study the characteristics and functions of different types of germ cells and somatic cells in testicular tissue, it is need to use more advanced single-cell sequencing technology. This method can explore in detail the specific role of various cells in testicular tissue in spermatozoon development and reveal the interactions and molecular mechanisms between different cell types. 

In addition, since spermatogenesis takes a long time, and this study only cultured in vitro for 7 days, the culture time is relatively short. This is a relative limitation of this study and it is noted in the text. 

Reviewer 4 Report

Comments and Suggestions for Authors

This paper deals with an undoubtedly interesting topic with major implications on practical andrology. Any progress in the preservation of testicular tissue with prospects to improve the reproductive potential of vulnerable patients is highly welcome in assisted reproduction.

The theme of the paper is very interesting, and I appreciate an unconventional approach, especially when human samples are available. Nevertheless, the manuscript is not well written and does not deliver a comprehensive interpretation of the collected data.

First of all, the abstract does not fully inform on the methodology and the results section is just composed of a list of differentially expressed genes. The background informing on an in vitro model of ovarian tissue is not relevant.

I am missing a rationale for the study in the introduction section. The paragraphs comprising the introduction are disconnected and are missing a logical interconnection. If the aim of the study was to evaluate two thawing regimens, why agarose and microfluidics-based systems are mentioned? What system was used in this study?

More specifics on the samples are needed: How many samples were collected from azoospermic patients and how many were obtained from cancer patients prior to treatment? Were there any exclusion criteria for the sample collection? Were the oncology patients azoospermic as well? If 6 samples were available and 4 groups were established with 3 samples were used per experimental group, how were the samples, how and based on what system were the samples distributed amongst the groups?

What exactly was evaluated in the Histology experiments? How was histology evaluated? What was the instrumentation (Microscope? A photomicrograph or morphometry program?)

Subsections 2.5. and 2.6. are missing references. Either provide them or the methodological subsections need more detail to assure reproducibility of the experiments.

Morphological changes or differences are not described and interpreted at all in the results section. Arrows or asterixis should be included in the photomicrographs to indicate any changes in the stricture with a proper description of the differences observed amongst the groups.

The Discussion section is almost non-existent. Subsection 4.1.is essentially a repetition of the aims, methodology and results in present tense, with only 3 references leading us back to papers previously published by the authors, two of which focus on ovarian tissue. Subsections 4.2. and 4.3. essentially name the differentially expressed genes and their function, without any implications these genes may play in the dynamics of testicular tissue in this study or comparisons with previously published reports. The discussion culminates in an essentially copied portion of a previous article done on ovarian tissue reciting the methodology and some collected data which have no connection or importance for this study whatsoever.

 The conclusion section simply states that a quick thawing (by 100°C) with following encapsulation in fibrin and long-time in vitro culture in a large volume of culture medium with constant stirring is a more suitable approach for testicular tissue cryopreservation. However, is unclear why since there is no proper explanation or interpretation of the collected data.

Limitations of the study are missing.

References are not formatted according to the Instructions for Authors.

In addition,  just one of these focuses on testicular tissue that is the primary goal of the paper I have reviewed. The rest of them deal with ovarian tissue which is unrelated to the topic of the paper, and whist I may accept reference number 9 to serve as a guiding line for the experimental design of this paper, and  references 17, 18 and 20 are cited in the methodological section (although without explanation what are the differences in the approach to the freezing of two different tissues - references 17 and 18 are related to ovarian tissue and reference 20 to testicular tissue), information of the reference 46 is completely unnecessary for the discussion section for it is a mere summary of a previous study without any value for the outcomes of the current paper.

REF. 9 Zhou, Y., W. Wang, P. Todorov, C. Pei, E. Isachenko, G. Rahimi, P. Mallmann, F. Nawroth, and V. Isachenko. "Rna Transcripts  in Human Ovarian Cells: Two-Time Cryopreservation Does Not Affect Developmental Potential." Int J Mol Sci 24, no. 8 (2023).  

REF. 17 Isachenko, V., B. Morgenstern, P. Todorov, E. Isachenko, P. Mallmann, B. Hanstein, and G. Rahimi. "Patient with Ovarian Insufficiency: Baby Born after Anticancer Therapy and Re-Transplantation of Cryopreserved Ovarian Tissue." J Ovarian Res 13, no. 1 (2020): 118.   

REF. 18 Isachenko, V., P. Todorov, E. Isachenko, G. Rahimi, B. Hanstein, M. Salama, P. Mallmann, A. Tchorbanov, P. Hardiman, N. Getreu, and M. Merzenich. "Cryopreservation and Xenografting of Human Ovarian Fragments: Medulla Decreases the Phosphatidylserine Translocation Rate." Reprod Biol Endocrinol 14, no. 1 (2016): 79.   

REF. 20 Pei, C., P. Todorov, M. Cao, Q. Kong, E. Isachenko, G. Rahimi, N. Mallmann-Gottschalk, P. Uribe, R. Sanchez, and V. Isachenko.  "Comparative Transcriptomic Analyses for the Optimization of Thawing Regimes During Conventional Cryopreservation of Mature and Immature Human Testicular Tissue." Int J Mol Sci 25, no. 1 (2023).   

REF. 46 Isachenko, V., I. Lapidus, E. Isachenko, A. Krivokharchenko, R. Kreienberg, M. Woriedh, M. Bader, and J. M. Weiss. "Human  Ovarian Tissue Vitrification Versus Conventional Freezing: Morphological, Endocrinological, and Molecular Biological Evaluation." Reproduction 138, no. 2 (2009): 319-27.

Comments on the Quality of English Language

The Results and Discussion section are written in present tense which is quite uncommon and not appropriate.

Author Response

Reviewer 4.

This paper deals with an undoubtedly interesting topic with major implications on practical andrology. Any progress in the preservation of testicular tissue with prospects to improve the reproductive potential of vulnerable patients is highly welcome in assisted reproduction.

The theme of the paper is very interesting, and I appreciate an unconventional approach, especially when human samples are available. .

Authors.

Dear Sir/Madam,

Thank you very much for your careful work with our manuscript. We accept all your remarks and respective changes were introduced in the text.

Reviewer 4. First of all, the abstract does not fully inform on the methodology and the results section is just composed of a list of differentially expressed genes. The background informing on an in vitro model of ovarian tissue is not relevant.

Autors.

Dear Sir/Madam,

Your remark is accepted and now we have written the following:

Background: Earlier it was established that 7 days in vitro culture of human ovarian tissue with the soluble extract of basement membrane protein (Matrigel) in 3-D culture system leads to a rise in gene fusion and alternative splicing events, potentially affecting gene expression and cellular functions. This method cannot be used for in vitro culture of human testicular tissue.

Reviewer 4. I am missing a rationale for the study in the introduction section. The paragraphs comprising the introduction are disconnected and are missing a logical interconnection. If the aim of the study was to evaluate two thawing regimens, why agarose and microfluidics-based systems are mentioned? What system was used in this study?

Autors.

Dear Sir/Madam, your remark is accepted and now we have re-formed structure of Introduction:

  1. In vito culture of testicular tissue is actual.
  2. There are different methods of such in vitro culture which involves a culture in small volumes of medium.
  3. One from these methods (probably most popular) we have tested and concluded that it is bad to due different genetic alterations.
  4. The aim of our investigations was to test new method of in vitro culture.

Reviewer 4. More specifics on the samples are needed: How many samples were collected from azoospermic patients and how many were obtained from cancer patients prior to treatment? Were there any exclusion criteria for the sample collection? Were the oncology patients azoospermic as well? If 6 samples were available and 4 groups were established with 3 samples were used per experimental group, how were the samples, how and based on what system were the samples distributed amongst the groups?

Autors.

Dear Sir/Madam, your remark is accepted and now we have written the following:

The technology being tested in our studies is being developed for cancer patients. However, the testicular tissue fragments used for the experiments were obtained from patients from a fertility treatment program. Five patients were diagnosed with obstructive azoospermia, one patient was diagnosed with restricted azoospermia. The informed consent was obtained from patients whose tissue was collected for this study. Testicular tissue fragments (3 to 4 mm3) was obtained from 6 patients aged from 32 to 42 (median age 36,5 years). Two tissue fragments from each patient were used for experiments (total 12 tissue samples were collected for research.

Reviewer 4. What exactly was evaluated in the Histology experiments? How was histology evaluated? What was the instrumentation (Microscope? A photomicrograph or morphometry program?)

Subsections 2.5. and 2.6. are missing references. Either provide them or the methodological subsections need more detail to assure reproducibility of the experiments.

Autors.

Dear Sir/Madam,

Thank you very much for your suggestion. The histological experiment is to observe the morphology of testicular tissue after different protocols. Now it is written:

The section was subjected to morphological analysis of testicular tissue development and viability under a Nikon Diaphot 300 microscope (Nikon, Tokyo, Japan).

According to your suggestion, we have added relevant references in Subsections 2.5. and 2.6:

2.5. Histology

Testicular tissues were fixed in 3.5% paraformaldehyde for 12h at 4°C. The fragments were embedded in paraffin wax. Sections of 5µm thickness were prepared and every 10th section of each sample was mounted on glass slides and stained with hematoxylin and eosin[21]. The section was subjected to morphological analysis of testicular tissue development and viability under a Nikon Diaphot 300 microscope.

2.6. Data Extraction and Sequencing

Each testicular tissue sample underwent RNA extraction using the Trizol method. Strand-specific transcriptome library construction was achieved by enriching mRNA from total RNA, followed by sequencing using the DNBSEQ high-throughput platform and subsequent bioinformatics analysis. The library was validated using the Agilent Technologies 2100 bioanalyzer. The library was then amplified using phi29 DNA polymerase to produce DNA nanoballs (DNBs), each containing more than 300 copies of a single molecule. The DNBs were loaded onto a patterned nanoarray, and single-end 50-base (or paired-end 100/150-base) reads were generated using combinatorial Probe-Anchor Synthesis (cPAS)[19, 22].The RNA-seq analysis was executed using the Dr. Tom System (https://biosys.bgi.com). Download the results of KEGG and GO related enrichment through this system. The raw RNA-seq data is accessible under BioProject: PRJNA1030294 and can be downloaded from the "Sequence read archive" on the National Center for Biotechnology Information (https://www.ncbi.nlm.nih.gov/bioproject/1030294).

  1. Pei, C., P. Todorov, M. Cao, Q. Kong, E. Isachenko, G. Rahimi, N. Mallmann-Gottschalk, P. Uribe, R. Sanchez, and V. Isachenko. "Comparative Transcriptomic Analyses for the Optimization of Thawing Regimes During Conventional Cryopreservation of Mature and Immature Human Testicular Tissue." Int J Mol Sci 25, no. 1 (2023).
  2. Amelkina, O., A. M. D. Silva, A. R. Silva, and P. Comizzoli. "Transcriptome Dynamics in Developing Testes of Domestic Cats and Impact of Age on Tissue Resilience to Cryopreservation." BMC Genomics 22, no. 1 (2021): 847.
  3. Love, M. I., W. Huber, and S. Anders. "Moderated Estimation of Fold Change and Dispersion for Rna-Seq Data with Deseq2." Genome Biol 15, no. 12 (2014): 550.

Reviewer 4. Morphological changes or differences are not described and interpreted at all in the results section. Arrows or asterixis should be included in the photomicrographs to indicate any changes in the stricture with a proper description of the differences observed amongst the groups.

Autors.

Dear Sir/Madam,

Your remark is accepted and now we have modified Fig. 4 and written the following:

HE-staining results of testicular tissue showed that compared with group 1 (quick thawing), the gap between the basement membrane and cells of the seminiferous tubules in group 2 (quick thawing + in vitro culture) was reduced, the basement membrane rupture was improved, and the nuclear condensation of spermatogonia was reduced (Fig.4, A1, A2, B1, B2). Similarly, the observation results of group 4 (slow thawing + in vitro culture) compared with group 3 (slow thawing) alone showed the same trend (Fig.4, C1, C2, D1, D2). These results indicate that in vitro culture of testicular tissue after cryopreservation is helpful for the recovery of its function.

Reviewer 4. The conclusion section simply states that a quick thawing (by 100°C) with following encapsulation in fibrin and long-time in vitro culture in a large volume of culture medium with constant stirring is a more suitable approach for testicular tissue cryopreservation. However, is unclear why since there is no proper explanation or interpretation of the collected data.

Autors.

Dear Sir/Madam,

Your remark is accepted and now we have written in Abstract as well as in Conclusion the following:

Quick thawing of human testicular cells (by 100°C), following encapsulation in fibrin and long-time in vitro culture in a large volume of culture medium with constant stirring do not cause disturbances in RNA and genes expressions.

Reviewer 4. Limitations of the study are missing.

Autors.

Your remark is accepted and now we have:

Reviewer 4. References are not formatted according to the Instructions for Authors.

Autors.

Dear Sir/Madam,

Your remark is accepted and it is corrected.

Reviewer 4. In addition,  just one of these focuses on testicular tissue that is the primary goal of the paper I have reviewed. The rest of them deal with ovarian tissue which is unrelated to the topic of the paper, and whist I may accept reference number 9 to serve as a guiding line for the experimental design of this paper, and  references 17, 18 and 20 are cited in the methodological section (although without explanation what are the differences in the approach to the freezing of two different tissues - references 17 and 18 are related to ovarian tissue and reference 20 to testicular tissue), information of the reference 46 is completely unnecessary for the discussion section for it is a mere summary of a previous study without any value for the outcomes of the current paper.

Dear Sir/Madam,

Your remark is accepted and now we have written the following (in part of Discussion regarding differences in cryopreservation of testicular and ovarian tissues):

In our experiments, the protocols used for cryopreservation of testicular tissue were used, which were previously used for ovarian tissue (9, 17, 18,20).

In our opinion, when we write about cryopreservation of ovarian and testicular tissue, the cells of both tissues can be presented as similar objects. In ovarian, as well as in testicular fragments, seven types of cells with similar intracellular structures can be differentiated. The main difference between testicular and ovarian tissues is their different density. The density of ovarian tissue is high, and testicular tissue has a loose structure. That is why for in vitro cultivation of ovarian fragments with a constantly stirred medium, there is no need to use encapsulation of this tissue. Whereas due to the looseness of testicular tissue, after several days of in vitro cultivation in a large volume, with stirring of the medium, a fragment of such tissue will be dispersed into suspension with small fragments. Therefore, we used fibrin capsules for cultivation of testicular tissue. However, the above differences in tissue density are insignificant in essence for cryopreservation: the cytoplasmic structures of cells of both tissues will be saturated with permeable cryoprotectants within minutes of the start of equilibration.

Authors:

Again thank you for your work.

 .

Round 2

Reviewer 1 Report

Comments and Suggestions for Authors

As reported before, manuscript is well written and subject related to the 3D in vitro culture of testicular tissues from human patients is very interesting. In general, authors addressed all our previous suggestions. At this point, I have no further considerations and, in my opinion, manuscript is able to be accepted at the present form. 

Author Response

Answer to reviewers.

Reviewer 1

As reported before, manuscript is well written and subject related to the 3D in vitro culture of testicular tissues from human patients is very interesting. In general, authors addressed all our previous suggestions. At this point, I have no further considerations and, in my opinion, manuscript is able to be accepted at the present form. 

Authors: Thank you very much for reviewing this article again in your busy schedule.

Reviewer 3 Report

Comments and Suggestions for Authors

The authors did not provide any new data that the reviewer suggested in the previous version. The authors just explain why they cannot provide the information. The explanations of the authors are not acceptable.

Author Response

Reviewer 3

The authors did not provide any new data that the reviewer suggested in the previous version. The authors just explain why they cannot provide the information. The explanations of the authors are not acceptable.

Authors: Thank you very much for reviewing this article again.

Regarding the patient's age, type of azoospermia, size of the testis. I have already written in 2.2. Collection of samples. “Testicular tissue fragments (3 to 12 mm3) were obtained from 6 patients aged from 32 to 42 (median age 36.5 years). Two tissue fragments from each patient were used for ex-periments (total 12 tissue samples were collected for research.”

“About the type of somatic cells and the types of spermatogenic cells present before and after freezing and after thawing and culture. and data of correlation between RNA seq and the presence/activity of testicular somatic cells.”

This study did not analyze the various types of testicular cells before and after freezing, which is a shortcoming of the study. However, due to the scarcity of samples, it is very difficult to collect samples in a short period of time and then analyze various cell types before and after freezing. I am sorry.

RNA sequencing is only a general study of the differential gene expression of different types of tissues. This type of sequencing cannot more accurately distinguish different cell types and changes. If I want to specifically study the relevant data of different testicular cells, I need to perform single-cell sequencing, which is a more advanced sequencing. So, regarding your question, if I want to obtain relevant data of different testicular cells, I need to re-perform single cell sequencing as a whole.

Reviewer 4 Report

Comments and Suggestions for Authors

The manuscript has undergone a robust round of revisions and I appreciate the efforts invested by the authors to edit the paper and to address my comments.

There are still several issues that require attention.

In the Introduction section, the authors once again provide a complete protocol for in vitro culture of cryopreserved human ovarian tissue. This was moved from the Discussion section, and it is not relevant for the purpouses of this paper. If readers are interested into the full details of the experimental approach on ovarian tissue, they will find a complete protocol in the referenced studies. I highly recommend to delete the details listed in Lines 82-95 alltogether.

Text written in Lines 96-105 and Discussion section connected to the obtained data should be written in past, not present tense. 

Several section of the Discussion are missing references to previous papers the authors are discussing with (Lines 330-353, Lines 362-379, Lines 388-391).

The Conclusion section needs an addition to the statement stemming from the data. The authors conclude that “Quick thawing of human testicular cells (by 100°C), following encapsulation in fibrin and long-time in vitro culture in a large volume of culture medium with constant stirring do not cause disturbances in DNA and genes expressions.” So, what would be its practical application?

Comments on the Quality of English Language

The issue with writing in present tense as opposed to past tense particularly in the Results and Discussion section is still present.

Author Response

Reviewer 4

In the Introduction section, the authors once again provide a complete protocol for in vitro culture of cryopreserved human ovarian tissue. This was moved from the Discussion section, and it is not relevant for the purpouses of this paper. If readers are interested into the full details of the experimental approach on ovarian tissue, they will find a complete protocol in the referenced studies. I highly recommend to delete the details listed in Lines 82-95 alltogether.

Authors: Thank you very much for your suggestions. Considering your highly recommend, we have deleted this part.

Text written in Lines 96-105 and Discussion section connected to the obtained data should be written in past, not present tense. 

Authors: Thank you very much for your suggestions. We have completed a careful examination of the tense of the text.

Several section of the Discussion are missing references to previous papers the authors are discussing with (Lines 330-353, Lines 362-379, Lines 388-391).

Authors: Thank you very much. Lines 330-353 are figure legend, Lines 362-379, Lines 388-391 are the results section of the article, so no relevant references are cited.

The Conclusion section needs an addition to the statement stemming from the data. The authors conclude that “Quick thawing of human testicular cells (by 100°C), following encapsulation in fibrin and long-time in vitro culture in a large volume of culture medium with constant stirring do not cause disturbances in DNA and genes expressions.” So, what would be its practical application?

Authors: Thank you very much for your suggestions. We rewrote the conclusion again.

“Encapsulation of testicular tissue in fibrin and long-time in vitro culture with constant stirring in a large volume of culture medium can reduce the impact of different thawing methods on cryopreserved testicular tissue. In vitro culture is very effective for spermatogenesis in cryopreserved testicular tissue, and it is recommended to be widely used in clinical medicine.”

The issue with writing in present tense as opposed to past tense particularly in the Results and Discussion section is still present.

Authors: Thank you very much for your suggestions. We have carefully checked the tense issues of the entire manuscript (especially the Results and Discussion sections).